# Personalized Management for Heart Failure with Preserved Ejection Fraction

**DOI:** 10.3390/jpm13050746

**Published:** 2023-04-27

**Authors:** Chang-Yi Lin, Heng-You Sung, Ying-Ju Chen, Hung-I. Yeh, Charles Jia-Yin Hou, Cheng-Ting Tsai, Chung-Lieh Hung

**Affiliations:** 1Division of Cardiology, Department of Internal Medicine, MacKay Memorial Hospital, No. 92, Sec. 2, Zhongshan N. Road, Taipei 10449, Taiwan; b101099136@tmu.edu.tw (C.-Y.L.);; 2Telemedicine Center, MacKay Memorial Hospital, Taipei 10449, Taiwan; 3Departments of Internal Medicine, Mackay Medical College, New Taipei City 25245, Taiwan; 4Mackay Junior College of Medicine, Nursing and Management, New Taipei City 25245, Taiwan; 5Institute of Biomedical Sciences, Mackay Medical College, New Taipei City 25245, Taiwan

**Keywords:** heart failure with preserved ejection fraction, artificial intelligence, phenotype, cluster, machine learning, latent class analysis

## Abstract

Heart failure with preserved ejection fraction (HFpEF) is a heterogeneous clinical syndrome with multiple underlying mechanisms and comorbidities that leads to a variety of clinical phenotypes. The identification and characterization of these phenotypes are essential for better understanding the precise pathophysiology of HFpEF, identifying appropriate treatment strategies, and improving patient outcomes. Despite accumulating data showing the potentiality of artificial intelligence (AI)-based phenotyping using clinical, biomarker, and imaging information from multiple dimensions in HFpEF management, contemporary guidelines and consensus do not incorporate these in daily practice. In the future, further studies are required to authenticate and substantiate these findings in order to establish a more standardized approach for clinical implementation.

## 1. Introduction

Heart failure with preserved ejection fraction (HFpEF) is diagnosed as heart failure with a left ventricular ejection fraction (LVEF) of ≥50% and elevated left ventricular filling pressures at rest or during exercise after careful exclusion of conditions that may mimic HFpEF [1,2]. HFpEF is a complex clinical syndrome that differs from other cardiovascular diseases, as it is defined by a combination of symptoms, signs, and other manifestations rather than a specific diagnostic test.

There is currently little evidence supporting the effectiveness of conventional therapies utilized for HFpEF to reduce mortality rates, such as empagliflozin in the EMPEROR-Preserved trial and dapagliflozin in the DELIVER trial. However, emerging research suggests that treatment should be tailored to the specific comorbidities present in each patient [3]. Some of the most common comorbidities seen in patients with heart failure include coronary artery disease, atrial fibrillation (AF), obesity, diabetes, renal impairment, and pulmonary hypertension. Accordingly, HFpEF can be classified into different phenotypes based on various criteria, including underlying etiology, clinical characteristics, and comorbidities [3,4,5,6,7,8]. Detailed molecular signaling, gene ontology functional analysis, and the use of the Kyoto Encyclopedia of Genes and Genomes pathway also potentiate the precise mechanisms of action and targets of SGLT2 inhibitors in clinical practice [9,10]. A comprehensive understanding and the specific pathways identified from HFpEF phenotyping also facilitate animal experimental studies to address relevant pathophysiological signaling [11].

## 2. Clinical Entities

Several studies have examined the relationship between HFpEF and clinical entities and comorbidities. For example, in the TOPCAT trial, it was discovered that patients with HFpEF who had been previously hospitalized for heart failure exhibited a greater likelihood of experiencing cardiovascular death, heart failure hospitalization, or aborted cardiac arrest than their counterparts who did not have a history of hospitalization [12]. The utilization of clustering methods shows great potential in addressing the heterogeneity of HFpEF and uncovering sub-phenotypes. Unbiased clustering methods have been used in recent studies to categorize distinct phenotypes among patients with HFpEF, taking into account their clinical characteristics, echocardiographic observations, and biomarker concentrations (Table 1).

Since the amount of healthcare data generated on a daily basis is overwhelming for a contemporary doctor, far surpassing the computational capacity of the human brain [25], machine learning (ML) and artificial intelligence (AI), which may use interchangeably, are increasingly being used in medical research to identify sub-phenotypes of diseases such as HFpEF. Artificial intelligence (AI) techniques are increasingly being used in medical research to identify sub-phenotypes of diseases such as HFpEF. These methods utilize statistical algorithms to analyze complex relationships between various patient characteristics and create distinct clusters that define sub-phenotypes of the disease. One technique is supervised learning, where an algorithm is trained on a pre-labeled dataset to predict outcomes based on new data. For example, machine learning (ML), does not rely on pre-labeled data and instead uses algorithms to identify patterns and relationships within the data. Latent class analysis (LCA) is a model-based clustering technique used in medical research to identify sub-phenotypes of diseases. LCA relies on a probabilistic model to describe the distribution of data, which is used to derive clusters from the data based on the probabilities that certain cases belong to certain latent classes, and relies less on a distance measure to find the clusters. Overall, these techniques offer promising tools and research approaches to identify sub-phenotypes of diseases such as HFpEF and can help to improve diagnosis, treatment, and prognosis from multiple dimensions, including clinical data, biomarkers, or imaging studies [7,13,23,26,27,28,29,30,31,32]. Herein, we listed several key landmark studies unraveling AI-based learning and phenotyping among the HFpEF population.

One method to subtype HFpEF is based on the presence of underlying clinical entities or comorbidities that contribute to the development and progression of the disease, known as clinical phenotyping (Figure 1A) [8]. The identification of specific clinical entities and comorbidities associated with different HFpEF phenotypes may have important clinical implications, as it may help guide the development of targeted therapies for these subtypes. For example, patients with HFpEF and pulmonary vascular disease may benefit from therapies that target pulmonary hypertension, whereas patients with HFpEF and metabolic dysfunction may benefit from weight loss and metabolic management strategies. Further, it has been proposed that certain natriuretic peptide deficiency syndrome may exist in HFpEF (e.g., obesity related HFpEF) [33]. Despite experiencing abnormally high left ventricular filling pressure during exercise, patients with this condition may benefit from treatment with a Neprilysin inhibitor, such as Entresto/Sacubitril (ARNi) [34]. Taken collectively, the optimal management for HFpEF can be tailored and personalized, targeting precision medicine in clinical practice [13,29].

Hwang et al. explored the implications of coronary artery disease (CAD) in HFpEF. The authors suggested that HFpEF patients with CAD have distinct clinical and pathophysiological characteristics compared with those without CAD. CAD may play a significant role in the progression of HFpEF, and a better understanding of this relationship could lead to the development of more effective treatments. This study provides insights into the clinical entity and underlying comorbidity of HFpEF, and it highlights the importance of considering these factors when defining and classifying HFpEF phenotypes [29].

Shah et al. identified distinct subgroups of HFpEF patients based on their comorbidities, including obesity-related comorbidities, metabolic syndrome, and pulmonary hypertension. This study showed that these subtypes had distinct clinical, biochemical, and imaging profiles, suggesting that they may represent different disease processes. The authors proposed that this phenomapping model-based clustering approach could be used to tailor treatments to individual patients and to improve outcomes in HFpEF [13] (Figure 1B).

Borlaug et al. investigated the associations between specific clinical entities, comorbidities, and HFpEF subtypes. Their study included 344 patients with HFpEF who were classified into four subtypes, based on the clinical entities or comorbidities present, as follows: (1) obesity-related, (2) hypertension-related, (3) diabetes-related, and (4) idiopathic. According to this research, the HFpEF subtype associated with obesity showed a greater prevalence of metabolic risk factors and more severe diastolic dysfunction compared to the other subtypes. The hypertension-related HFpEF subtype had a higher prevalence of left ventricular hypertrophy, concentric remodeling, and worse systolic function. The diabetes-related HFpEF subtype had a higher prevalence of diabetic nephropathy, worse systolic function, and more severe diastolic dysfunction. The idiopathic HFpEF subtype had a lower prevalence of comorbidities and less severe diastolic dysfunction than the other subtypes [8].

Uijl et al. proposed a five-cluster model, labeled Cluster 1 and Clusters 2–5, among 6909 HFpEF from the Swedish Heart Failure Registry (SwedeHF) and externally validated this in 2153 patients from the Chronic Heart Failure ESC-guideline based Cardiology practice Quality project (CHECK-HF) registry, as shown in Figure 2. Cluster 1 included young patients with low comorbidity burdens and the highest proportion of implantable devices. Cluster 2 included patients with AF and hypertension without diabetes. Cluster 3 included the oldest patients with the most cardiovascular comorbidities. Cluster 4 included patients with obesity, diabetes, and hypertension. Cluster 5 included older patients with ischemic heart disease, hypertension, and renal failure; these patients were most frequently prescribed diuretics. With this clustering, patients in the same cluster may have more homogeneity, which may contribute to more beneficial medical therapies. Patients in the young–low comorbidity burden cluster had the lowest event rates, while patients in the older–AF and cardio–renal clusters had the highest event rates [23].

## 3. Imaging

Cardiac imaging and measurement of the cardiac structure are critical aspects in diagnosing HFpEF, as the symptoms of this condition can be nonspecific. Non-invasive measures of cardiac structure and function can assist in improving diagnostic accuracy and differentiating the sub-phenotypes of HFpEF. Additionally, imaging can exclude alternative diagnoses that mimic HFpEF, such as hypertrophic cardiomyopathy, primary valvular heart disease, cardiac amyloidosis, and pericardial disease. While two-dimensional (2-D) transthoracic echocardiography is the most commonly used imaging modality, advanced imaging techniques, including cardiac magnetic resonance imaging and 2-D speckle tracking echocardiography, are used to identify distinct HFpEF phenotypes based on left ventricular structure and function [13,23,28,29,30,31,32,35]. Compared with standard 2-D echocardiography, three-dimensional (3-D) echocardiography provides a more reliable and reproducible evaluation of cardiac chamber volumes, mass, and shape, which are highly correlated with cardiac magnetic resonance imaging (CMR) [36]. CMR provides comprehensive information on morphology, function, perfusion, viability, and tissue characteristics. CMR can detect fibrosis, lipid content, and energy metabolism, making it a valuable tool for assessing suspected CAD and detecting coronary microvascular disease (CMD) in the future [37].

Several studies have investigated the use of echocardiography to identify the imaging-based phenotypes of HFpEF. Diastolic dysfunction, a hallmark feature of HFpEF, has been well adopted in the initial classification of HFpEF [38]. One study used a combination of 2-D speckle-tracking echocardiography and CMR to identify three distinct phenotypes based on left ventricular structure and function [38]. These phenotypes included left ventricular hypertrophy with preserved global longitudinal strain (GLS), left atrial enlargement with normal GLS, and normal left ventricular and left atrial structures with impaired GLS. The aforementioned study found that these phenotypes were associated with different clinical and biochemical features and may have different prognostic implications [23,35,38].

Another study evaluated left atrial function using 2-D speckle tracking echocardiography and found that left atrial strain was significantly decreased in patients with HFpEF compared with that in controls. The study also found that left atrial strain was independently associated with a higher risk of adverse events, including hospitalization and mortality [39].

Cardiac imaging and measures of cardiac structure and function are essential for diagnosing HFpEF, excluding alternative diagnoses, and identifying imaging-based phenotypes. Although 2-D echocardiography remains the most commonly used imaging modality, CMR and 3-D echocardiography offer more comprehensive information regarding cardiac structure and function. These imaging techniques may have important prognostic implications and help guide management strategies in patients with HFpEF.

## 4. Management of HFpEF Phenotype Based on “SwedeHF” and “CHECK-HF” Registries

Personalized management of different HFpEF phenotypes using clustering targeting more specific molecular or pathological etiology driving underlying mechanisms has been proposed in several studies [27,40,41,42]. For example, obesity-related HFpEF with or without hyperlipidemia or diabetes may benefit from combined sodium–glucose cotransporter-2 inhibitors (SGLT2i), mineralocorticoid receptor antagonists (MRA), and angiotensin receptors/neprilysin inhibitor (ARNi) due to an inner deficiency of effective natriuretic peptide from excessive visceral adiposity [43]. Herein, we provided an example of the possible therapeutic implications of performing phenotyping among the HFpEF population using findings from the “SwedeHF” and “CHECK-HF” registries [23]. Despite being promising, these studies may warrant further external validations that can be applicable across different races with wide clinical settings to show how these findings can be interpreted practically and implemented from the working hypotheses (Table 2). Thus, these findings and their implications should be discussed in the broadest context possible.

### 4.1. Cluster 1

Among the five clusters, patients with HFpEF in this group had a median age of 59 years and a relatively low burden of comorbidities, making them the youngest of the cohorts. The most common comorbidities in Cluster 1 were hypertension (46%) and obesity (42%). The principles of management for this group are to control blood pressure and reduce body weight. It is worth mentioning that cluster 1 includes patients who have recovered HFrEF, due to the higher percentage of implantable cardioverter-defibrillator or cardiac resynchronization therapy.

In addition to the implantable devices, quite a few medications have demonstrated an established efficacy in previous HFpEF trials. Those drugs were renin–angiotensin–aldosterone system (RAAS) antagonists such as angioten-sin-converting enzyme inhibitors (ACEis), angiotensin II receptor blockers (ARBs), mineralocorticoid receptor antagonists (MRAs), and angiotensin receptors/neprilysin inhibitor (ARNi), which could be considered as first-line agents for the management of HFpEF. Lifestyle modifications were strongly suggested for this cluster. Significant improvements in quality of life and exercise tolerance were observed as a result of weight reduction, which was found to be safe. In addition to these benefits, weight loss in patients with HFpEF has been shown to have a positive impact on cardiac function and metabolic parameters, potentially leading to reduced doses of diuretics, antihypertensive agents, and diabetes medications.

### 4.2. Cluster 2

The individuals belonging to Cluster 2 were relatively older compared to those in Cluster 1, having a median age of 77 years. This cluster included patients with HFpEF characterized by AF without diabetes. Principles of management for this cluster align with the AF Better Care (ABC) pathway, including rate/rhythm control in AF management, as follows: (A) avoiding thromboembolic events with the use of anticoagulation. (B) better management of symptoms with personalized, symptom-directed decisions on rate or rhythm control. Rate control involves the use of beta-blockers/non-dihydropyridine (DHP) calcium channel blockers (CCBs) (diltiazem or verapamil)/digoxin; rhythm control involves the use of amiodarone/dronedarone or AF ablation. (C) Effective management of cardiovascular and coexisting conditions, including attention to psychological factors and lifestyle. Following the ABC pathway has been shown to lead to improved outcomes, including decreased risks of all-cause mortality, cardiovascular mortality, stroke, and hospitalization due to cardiovascular reasons. It is important to avoid excessive rate control in patients with both HFpEF and AF, as it may diminish their chronotropic reserve. In a trial comparing strict (<80 bpm) and lenient (<110 bpm) rate control in patients with AF, which may have included individuals with undiagnosed HFpEF, no significant differences in outcomes were observed.

### 4.3. Cluster 3

Among the five clusters, Cluster 3 patients were the oldest (median age, 88 years) with the highest N-terminal pro b-type natriuretic peptide (NT-proBNP) values. It was reasonable to eliminate any meaningful clinical phenotyping for this cluster, as it presented with an anticipated higher risk for an ominous outcome. Clinically, the principle of for the elderly is to reduce hospitalization rates and improve quality of life. Decongestion of diuretics has been shown to reduce hospitalization rates. In the TOPCAT trial, spironolactone was associated with a decrease in heart failure hospitalization rates compared with the placebo [12]. This cluster can be effectively managed with measures such as controlling heart rate in patients with AF, optimizing blood pressure control, and implementing lifestyle interventions such as exercise training to enhance functional capacity. When life comes to an end, palliative care, including symptom management and psychological, emotional, and spiritual support, should be properly offered to patients and caregivers throughout the disease course, not only in advanced stages.

### 4.4. Cluster 4

Cluster 4 was composed of patients who had a median age of 71 years and were identified as having diabetes but not AF. Serum glucose control is the mainstay of this cluster. SGLT2 inhibitors have emerged as a critical component of HFrEF therapy as they possess favorable pleiotropic effects on various body parts such as the kidney, liver, pancreas, blood vessels, and adipose tissue, apart from their primary role as an antidiabetic medication. The EMPEROR-Preserved trial was groundbreaking in the study of HFpEF as it compared the effects of empagliflozin with a placebo in patients with ejection fractions above 40%, irrespective of whether they had diabetes or not. The trial demonstrated a significant reduction in the risk of heart failure-related hospitalizations and cardiovascular mortality, as well as an improvement in renal outcomes.

According to the DELIVER trial, dapagliflozin is superior to placebos in decreasing cardiovascular deaths and hospitalizations due to heart failure in patients with mildly reduced or preserved ejection fractions. Additionally, the study showed that dapagliflozin was effective in patients who previously had ejection fractions below 40% but later saw an increase to over 40%.

Glucagon-like peptide-1 receptor agonists (GLP-1 RAs) are also associated with positive cardiovascular effects. A recent meta-analysis involving 592 patients revealed that liraglutide was connected with significant enhancements in the left ventricular diastolic function.

### 4.5. Cluster 5

Cluster 5 was a union of Clusters 2 and 4. This cluster had a median age of 82 years, and its members had comorbidities of both diabetes and AF. In the DECLARE-TIMI 58 trial (Multicenter Trial to Evaluate the Effect of Dapagliflozin on the Incidence of Cardiovascular Events), dapagliflozin reduced the incidence of AF in patients with diabetes [44]. Efforts for Clusters 2 and 4 should be applied to this cluster, including the ABC pathway for the management of AF and SGLT2i for diabetes.

## 5. Management of Obesity-Related HFpEF Phenotype

Obesity as a common etiology and co-morbidity for HFpEF has been shown to induce activated sympathetic system and RAAS (and thus hyperaldosteronism with sodium retention) and further promote systemic inflammation [45,46], which may subsequently augment impaired cardiac filling conditions and aggravate unfavorable cardiac remodeling and HF progression [47,48]. Hence, HFrEF patients with central obesity are particularly prone to therapeutic benefits with eplerenone use [49].

Elevated circulating levels of aldosterone, either directly from adipocytes or released from the adrenal gland in response to leptin through the adipokines-cell-signaling molecules secreted (from central obesity or visceral adipose tissue), along with the attenuated anti-aldosterone effects from natriuretic peptides due to an increased neprilysin activity in obesity, may potentiate [50] the deleterious effect of neprilysin HF patients with obesity regardless of HF phenotypes [33]. This “leptin-aldosterone-neprilysin axis activation”, when observed in part as natriuretic peptide deficiency syndrome, as observed in obesity-related HFpEF pathophysiology, may exacerbate the interaction of leptin and aldosterone to promote sodium retention, plasma volume expansion, and regional (such as myocardial) and systemic inflammation and fibrosis (Figure 3) [23,33,43,51].

Importantly, an activated leptin–aldosterone–neprilysin axis with sustained increases in aldosterone and neprilysin concentration may in turn accelerate the accumulation and inflammation of epicardial fat [52,53]. Recently, proteomics in the LIFE-Heart study (further verified in the Aldo-DHF validation cohort) targeting biomarkers involving volume expansion, myocardial fibrosis, and systemic inflammation has been shown to improve obesity-related HFpEF [54] phenotyping with a distinct biomarker signature. However, whether there may exist some clinical features (e.g., central obesity, region-specific adiposity, e.g., pericardial fat burden) with therapeutic implications using AI-assisted machine learning or clustering may warrant further research (Figure 3).

## 6. Conclusions

In summary, the identification and characterization of HFpEF phenotypes are important for guiding diagnosis, management, and research into novel treatment strategies. It was a prerequisite for us to identify that the presence of ischemic heart disease by itself induces prognostic implications. Given the complexity of HFpEF, a personalized approach to management that considers the underlying mechanisms and comorbidities in each patient is needed and might help to solve the puzzle of this challenging syndrome.

## Figures and Tables

**Figure 1 jpm-13-00746-f001:**
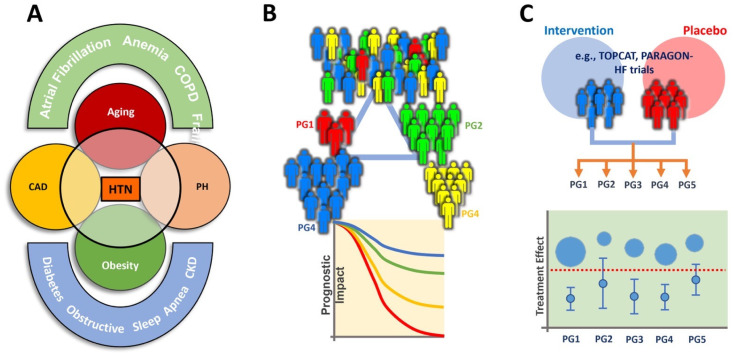
Clinical phenotyping of HFpEF (**A**), AI-assisted or latent class analysis (LCA) HFpEF phenotyping (**B**), and treatment-based clustering (**C**). PG: phenogroup. Other abbreviations are as in Table 1. Panel (**A**) was modified and adopted [8].

**Figure 2 jpm-13-00746-f002:**
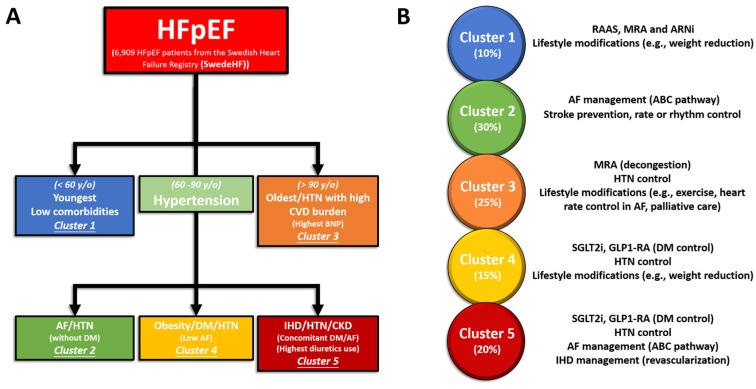
Schematic diagram showing characteristics for the 5 major phenotypic clusters of HFpEF from SwedeHF (*n* = 6909) (**A**) calculated with an LCA and externally validated in CHECK-HF (*n* = 2153). Percentage and treatment strategies of these HFpEF populations in the whole study cohort (**B**).

**Figure 3 jpm-13-00746-f003:**
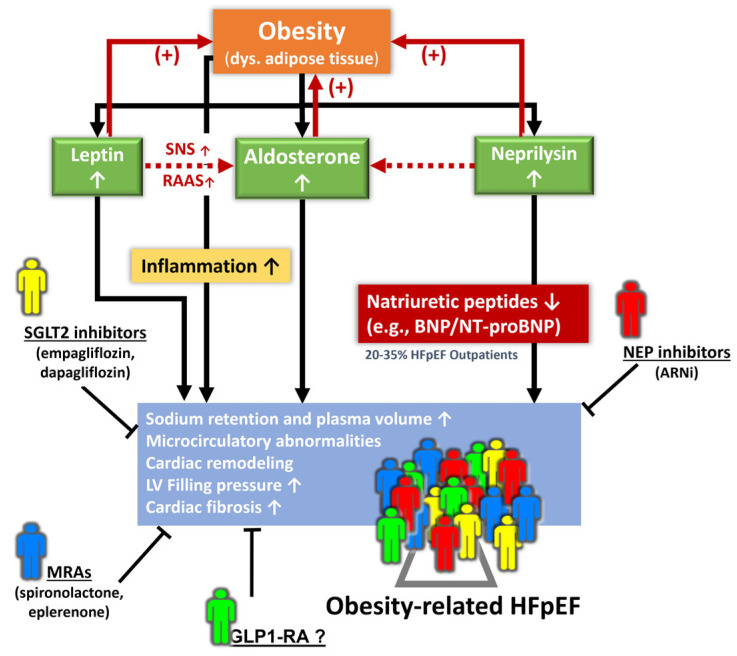
Pathophysiological signaling from the leptin–aldosterone–neprilysin axis activation underlying obesity-related HFpEF and potential diverse phenotypes for pharmacological interventions. ARNi: angiotensin receptor-neprilysin inhibitor; Dys.: dysregulated; GLP1-RA: glucagon-like peptide-1 receptor agonist; MRA: mineralocorticoid receptor antagonist; NEP: neprilysin; RAAS: renin–angiotensin–aldosterone system; SNS: sympathetic nervous system.

**Table 1 jpm-13-00746-t001:** Machine learning (ML) and latent class analysis (LCA) phenotyping of HFpEF.

**Machine-Learning**			
Study	Number of Subjects	Classification	Characteristics
Shah et al., 2015 [13]	397	Phenogroup 1	Natriuretic Peptide Deficiency Syndrome, young, obese, relatively fewer comorbidities
		Phenogroup 2	Extreme Cardiometabolic Syndrome, HTN, obesity (typically BMI > 35), DM
		Phenogroup 3	Right Ventricle-cardio-abdomino-renal Syndrome, CKD, PH, cardiorenal phenotype
Sanchez-Martinez et al., 2018 [14]	156	Cluster 1	Healthy cluster
		Cluster 2	HFpEF: Older, higher NTproBNP, BMI, impaired exercise tolerance at 6MWT, LV hypertrophy, higher E/e’ ratio
Przewlocka-Kosmala et al., 2019 [15]	228	Cluster 1	Normal CR/DR, normal increase in HR and diastolic function during exercise
		Cluster 2	Altered CR/DR, decreased exercise tolerance at CPET; chronotropic incompetence and diastolic dysfunction on exercise
Segar et al., 2020 [16]	654	Phenogroup 1	Older, several CV risk factors: obesity; DM, HTN, worse renal function, significant LV concentric remodeling, LA dilatation, diastolic dysfunction
		Phenogroup 2	Low prevalence of CV risk factors, moderate LV concentric remodeling, moderate LA dilatation, and higher prevalence of moderate MR
		Phenogroup 3	Intermediate burden of CV risk factors, mainly DM and HTN, moderate LV concentric remodeling and LA dilatation
Hedman et al., 2020 [17]	397	Phenogroup 1	HTN, IHD, DM, and CKD, marked LV concentric remodeling, modest electric remodeling (AF 37%)
		Phenogroup 2	Older age, HTN, significant LA dilatation and higher prevalence of RV failure, severe electric remodeling (AF 85%)
		Phenogroup 3	Younger, HTN, modest LV remodeling and electric remodeling (AF 48%)
		Phenogroup 4	HTN, significant LV and atrial remodeling, highest electrical remodeling (AF 90%)
		Phenogroup 5	HTN, IHD, moderate LV remodeling, moderate electrical remodeling (AF 43%)
		Phenogroup 6	Low BMI, severe LA remodeling, RV dysfunction; significant electric remodeling (AF 96%)
Schrub et al., 2020 [18]	356	Cluster 1	Younger, HTN, DM, obesity, CKD, less electric remodeling, LV hypertrophy, lowest rate of severe MR
		Cluster 2	Intermediate age, HTN, less LV remodeling, but significant LA atrial dilatation and higher severe MR rate
		Cluster 3	Oldest, severe electrical remodeling (AF 87%), severe LA dilatation, higher prevalence of severe MR
Woolley et al., 2021 [19]	429	Cluster 1	Highest frequency of CKD and DM
		Cluster 2	Elderly, high frequency of AF and HTN
		Cluster 3	Young, obese, fewest comorbidities
		Cluster 4	Highest rates of COPD, CAD, and smoking
Gu et al., 2021 [20]	970	Phenogroup 1	Relatively preserved NYHA class and few to no comorbidities
		Phenogroup 2	Higher proportion of women and prevalence of AF
		Phenogroup 3	Highest BMI, highest prevalence of IHD, DM, and severe symptoms assessed with NYHA
**Latent Class Analysis**			
Study	Number of Subjects	Classification	Characteristics
Kao et al., 2015 [21]	4113	Subgroup A	Median age 65, men, low rates of AF, CKD, valvular disease, and high rates of alcohol use
		Subgroup B	Median age 65, women, low rates of AF, CKD, valvular disease, and high rates of anemia
		Subgroup C	Median age 70, high rates of DM, obesity, HLD, CAD, CKD
		Subgroup D	Median age 73, women, average rates of DM, obesity, HLD, CKD
		Subgroup E	Median age 75, men, low BMI, high rates of AF, CAD
		Subgroup F	Median age 82, women, low BMI, high rates of AF, valvular disease, CKD and anemia
Cohen et al., 2020 [22]	3442	Phenogroup 1	Younger with mild symptoms lowest levels of NP, DM, CKD, and LV dysfunction, highest rates of smoking
		Phenogroup 2	Older with stiff arteries, small LVs and AF, women, highest rates of AF and CKD, low rates of obesity and DM
		Phenogroup 3	Obese diabetic with advanced symptoms, highest rates of obesity, DM, and high rates of CKD and depression
Uijl et al., 2021 [23]	6909	Cluster 1	Median age 59, more males, fewest comorbidities, most had NYHA class I/ll and normal eGFR
		Cluster 2	Median age 77, higher rates of AF and HTN, relatively normal eGFR and lowest rate of DM
		Cluster 3	Median age 88, more females, highest rate of AF, lowest BMI values
		Cluster 4	Median age 71 years, most likely male, higher BMI and almost all patients had HTN and DM
		Cluster 5	Median age 82, most likely female, higher BMI values and NYHA III/IV, IHD, AF, all patients had HTN and most had lower eGFR values

AF: atrial fibrillation, CAD: coronary artery disease, CKD: chronic kidney disease, COPD: chronic obstructive pulmonary disease, CPET: cardiopulmonary exercise test, DM: diabetes mellitus, HLD: hyperlipidemia, HTN: hypertension, IHD: ischemic heart disease, MR: mitral regurgitation, PH: pulmonary hypertension. Table was modified from reference number [24].

**Table 2 jpm-13-00746-t002:** Simple summary table for management of specific HFpEF phenotypes based on the “SwedeHF” and “CHECK-HF” registries.

Classification	Characteristics	Treatment Strategy
Cluster 1	Younger with low comorbidity	Lifestyle modifications Risk factor screening
Cluster 2	AF without T2DM	Restoration of normal sinus rhythm, anticoagulation, blood pressure control
Cluster 3	Oldest with many cardiovascular comorbidities	Diuretics, mineralocorticoid receptor antagonists, lifestyle interventions
Cluster 4	T2DM without AF	Glycemic control, SGLT2i
Cluster 5	T2DM and AF	SGLT2i

## Data Availability

Because of the review nature for this article, requests to access the dataset from qualified researchers are not applicable.

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
