# Peer review of "Personalized Management for Heart Failure with Preserved Ejection Fraction"

_jpm, 2023, doi:10.3390/jpm13050746_

Round 1

Reviewer 1 Report

The review is of clinical interest since it refers to the multifaced problem of HFpEF. The heterogeneity of the syndrome necessitates a more clinically meaningful process in therapeutic handling. Clustering might contribute to this providing specific phenotypes to be considered. The paper is useful since it presents in detail the existing contribution of cluster analysis.

A few remarks must be stated:

1.      The current review is based mainly in the findings of the study by Uijl et al. It has to be stated that the cluster 1 involves patients who have recovered HFrEF, due to the higher percentage implantable cardioverter-defibrillator or cardiac resynchronization therapy. Cluster 3 involves oldest patients (median age, 88 years) with an expected far increased risk for ominous outcome, thus eliminating any meaningful clinical phenotyping.

2.      Presence of ischemic heart disease by itself induces prognostic implications. Thus in the analysis concerning the role of HFpEF itself, it is a prerequisite to exclude patients with residual ischemic burden despite specific and appropriate treatment.

3.      The following paragraph is confusing and has to be omitted 

[Interestingly, Tsujimoto et al explored the treatment effect of resistant hypertension 124 in the TOPCAT trial (Figure 1C) [18]. Jackson et al explored the effect of ARNi in PARA- 125 GON-HF trial [19]. Recently, Karwath et al explored the therapeutic effect of beta-blocker 126 among 15,659 HFrEF (LVEF <50%) from 9 double blinded, randomized controlled trials 127 and identified several clusters belonged to either sinus rhythm or atrial fibrillation (AF) 128 population with diverse therapeutic effects (Figure 1C) [20]. Whether these observations 129 may be replicated among HFpEF may worth further research with future larger real- 130 world or pooled randomized trials data. ]

4.      The following paragraphs are of limited clinical value and has to be omitted 

[Furthermore, a study comparing 2-D and 3-D echocardiography in patients with 183 HFpEF found that 3-D echocardiography was more accurate in assessing left ventricular 184 mass and volume and was better able to identify left ventricular hypertrophy [30]. An- 185 other study found that SVI/S' is a non-invasive index calculated by three-dimensional and 186 tissue Doppler echocardiography as a substitute measure of pulmonary capillary wedge 187 pressure (PCWP) and can be used to diagnose and determine prognosis in cases of HFpEF 188 [31].]

[By using CT-based evaluation of LA geometric differences and 2-D speckle tracking 190 technique by echocardiography, and Kuo and Hung et al identified distinctive structural 191 and functional atrial phenotypes and visualized heatmap between HFpEF and AF in a 192 pilot study. This approach likely holds promise for future new phenotyping with poten- 193 tial for prognostic and therapeutic impacts among HFpEF population with or without 194 prevalent AF [32].]

5.      There is overemphasis of 3D echo. However, in clinical practice for HFpEf it is irrelevant and never being used.

Author Response

Response to Reviewer Comments

Reviwer 1

Point 1: The current review is based mainly in the findings of the study by Uijl et al. It has to be stated that the cluster 1 involves patients who have recovered HFrEF, due to the higher percentage implantable cardioverter-defibrillator or cardiac resynchronization therapy. Cluster 3 involves oldest patients (median age, 88 years) with an expected far increased risk for ominous outcome, thus eliminating any meaningful clinical phenotyping.

Response 1: We had stated that there are higher percentage ICD and CRT therapy in cluster 1 and an expected far increased risk for ominous outcome, thus eliminating any meaningful clinical phenotyping in cluster 3.

Point 2: Presence of ischemic heart disease by itself induces prognostic implications. Thus in the analysis concerning the role of HFpEF itself, it is a prerequisite to exclude patients with residual ischemic burden despite specific and appropriate treatment.

Response 2: We had added this point in the “conclusions”

Point 3: The following paragraph is confusing and has to be omitted

[Interestingly, Tsujimoto et al explored the treatment effect of resistant hypertension 124 in the TOPCAT trial (Figure 1C) [18]. Jackson et al explored the effect of ARNi in PARA- 125 GON-HF trial [19]. Recently, Karwath et al explored the therapeutic effect of beta-blocker 126 among 15,659 HFrEF (LVEF <50%) from 9 double blinded, randomized controlled trials 127 and identified several clusters belonged to either sinus rhythm or atrial fibrillation (AF) 128 population with diverse therapeutic effects (Figure 1C) [20]. Whether these observations 129 may be replicated among HFpEF may worth further research with future larger real- 130 world or pooled randomized trials data. ]

Response 3: Paragraph deleted

Point 4: The following paragraphs are of limited clinical value and has to be omitted

[Furthermore, a study comparing 2-D and 3-D echocardiography in patients with 183 HFpEF found that 3-D echocardiography was more accurate in assessing left ventricular 184 mass and volume and was better able to identify left ventricular hypertrophy [30]. An- 185 other study found that SVI/S' is a non-invasive index calculated by three-dimensional and 186 tissue Doppler echocardiography as a substitute measure of pulmonary capillary wedge 187 pressure (PCWP) and can be used to diagnose and determine prognosis in cases of HFpEF 188 [31].]

[By using CT-based evaluation of LA geometric differences and 2-D speckle tracking 190 technique by echocardiography, and Kuo and Hung et al identified distinctive structural 191 and functional atrial phenotypes and visualized heatmap between HFpEF and AF in a 192 pilot study. This approach likely holds promise for future new phenotyping with poten- 193 tial for prognostic and therapeutic impacts among HFpEF population with or without 194 prevalent AF [32].]

Response 4: These 2 paragraph was deleted

Point 5: There is overemphasis of 3D echo. However, in clinical practice for HFpEf it is irrelevant and never being used.

Response 5: Kindly received your point.

Reviewer 2 Report

Lin et al. Provide a very thoughtful review of HFpEF and it’s need for personalized management.

The review is well-written and consistent, finding some insights and thoughts to treat patients in a more personalized way in HFpEF.

All in all the graphics are nice, the paper well written and informative.

I have only minor comments.

-          Figure 1 C: please, write Placebo and Intervention group, the abbreviations are not needed.

-          I am not in agreement in calling late class analysis a novel algorithm which is not ML. The first chapter needs to be rewritten, since the authors seem to be somewhat confused in terms of ML and AI. I am well aware the distinction is difficult and some experts still are arguing, but calling LCA not ML and stating that ML is solely unsupervised is simply wrong. Please, consult other reviews (for example DOI: 10.1007/s00395-023-00982-7) and rewrite the first chapter “Clinical entities” with the correct definitions.

-          Fig. 1, caption: please rephrase. It is not either ML or LCA, rather LCA being a type of ML.

-          Introduction: "Empa preserved" and "Deliver" did show a mortality benefit in HFpEF. Please rephrase.

Author Response

Point 1: Figure 1 C: please, write Placebo and Intervention group, the abbreviations are not needed.

Response 1: Figure modified

Point 2: I am not in agreement in calling late class analysis a novel algorithm which is not ML. The first chapter needs to be rewritten, since the authors seem to be somewhat confused in terms of ML and AI. I am well aware the distinction is difficult and some experts still are arguing, but calling LCA not ML and stating that ML is solely unsupervised is simply wrong. Please, consult other reviews (for example DOI: 10.1007/s00395-023-00982-7) and rewrite the first chapter “Clinical entities” with the correct definitions.

Response 2: We thank the reivewer’s comment on this critical point. First chapter had been rewrite according to DOI: 10.1007/s00395-023-00982-7 and been cited with correct definitions of ML and AI. In particular, we omitted the description about LCA not belong to ML, and the sentence about ML as solely unsupervised.

We thank the reviewer’s opinion on this important clarification.

Point 3: Introduction: "Empa preserved" and "Deliver" did show a mortality benefit in HFpEF. Please rephrase.

Response 3: Rephrase of the paragraph was done.